# Identification and Characterization of Retinitis Pigmentosa in a Novel Mouse Model Caused by PDE6B-T592I

**DOI:** 10.3390/biomedicines11123173

**Published:** 2023-11-29

**Authors:** Chun-Hong Xia, Haiquan Liu, Mei Li, Haiwei Zhang, Xinfang Xing, Xiaohua Gong

**Affiliations:** Herbert Wertheim School of Optometry and Vision Science Program, University of California, Berkeley, CA 94720, USA; chxia@berkeley.edu (C.-H.X.); lhaiquan@yahoo.com (H.L.); mei.li@berkeley.edu (M.L.); haiweiwendy_zhang@berkeley.edu (H.Z.); shaunaxinfang1115@berkeley.edu (X.X.)

**Keywords:** PDE6B, retinal pigmentosa (RP), retinal degeneration (RD), recombinant adeno-associated virus (rAAV)

## Abstract

The cGMP-phosphodiesterase 6 beta subunit (PDE6B) is an essential component in the phototransduction pathway for light responses in photoreceptor cells. PDE6B gene mutations cause the death of rod photoreceptors, named as hereditary retinitis pigmentosa (RP) in humans and retinal degeneration (RD) in rodents. Here, we report a new RD model, identified from a phenotypic screen of N-ethyl-N-nitrosourea (ENU)-induced mutant mice, which displays retinal degeneration caused by a point mutation in the *Pde6b* gene that results in PDE6B-T592I mutant protein. The homozygous mutant mice show an extensive loss of rod photoreceptors at the age of 3 weeks; unexpectedly, the loss of rod photoreceptors can be partly rescued by dark rearing. Thus, this RD mutant model displays a light-dependent loss of rod photoreceptors. Both western blot and immunostaining results show very low level of mutant PDE6B-T592I protein in the retina. Structure modeling suggests that the T592I mutation probably affects the function and stability of PDE6B protein by changing intramolecular interactions. We further demonstrate that the expression of wild-type PDE6B delivered by subretinally injected adeno-associated virus (rAAV) prevents photoreceptor cell death in this RD model in vivo. The PDE6B-T592I mutant is, therefore, a valuable RD model for evaluating rAAV-mediated treatment and for investigating the molecular mechanism of light-dependent rod photoreceptor cell death that is related to impaired PDE6B function.

## 1. Introduction 

Retinitis pigmentosa (RP) is a genetically heterogeneous group of blinding diseases that affects millions of people worldwide; it is one of the leading causes of progressive vision loss and blindness in young individuals [1,2,3,4,5,6]. RP is the most common inherited retinal disease, caused by many different gene mutations that mostly affect rod photoreceptors in the retina. To date, there are over 100 genes have been identified to be associated with human RP (https://sph.uth.edu/retnet (accessed on 24 October 2023)). Characterized by a loss of rod photoreceptor cells and progressive vision loss, RP can be especially caused by impaired members in the phototransduction pathway. The β subunit of rod cGMP-phosphodiesterase type 6 (PDE6B) was the first causative gene mutation identified for RP [7,8]. PDE6B is a key component in the phototransduction cascades and plays an essential role in human vision [9]. RD mutant mice replicate human RP pathology and causative genetic mutations. Animal models of retinal degeneration (RD) provide valuable tools for understanding the underlying mechanisms for human RP and for testing potential new therapeutic approaches. There is a critical need for developing effective treatment to delay, prevent, or cure vision loss in human RP patients in clinics. Molecular pathology and efficacy of potential new treatments in mouse RD models can be effectively studied in a short time frame. The well-studied mouse *rd1* model, caused by the PDE6B-Y347X mutation, exhibits rapid retinal degeneration due to the extensive death of rods [10,11,12]. In the rod photoreceptor cells, PDE6B forms the PDE6 complex with PDE6α and PDE6γ subunits [13,14,15]. The inactive PDE6 in the dark allows cGMP to bind to cGMP gated ion channels, which keeps these cGMP bound channels open. Light absorption by rhodopsin triggers the phototransduction cascade. The PDE6 is activated by the release of the PDE6γ subunit from the PDE6αβ, causing the hydrolysis of cGMP; the ion channels close without the cGMP binding, leading to hyperpolarization of the cell membrane [13,16]. PDE6B-Y347X results in a nonfunctional PDE in the *rd1* mice, leading to increased cGMP and calcium levels in the photoreceptors [17,18] and, ultimately, the death of rod photoreceptor cells [10,11]. The *rd10* mice, caused by the PDE6B-R560C mutation, were later identified [19,20]. Mutant PDE6B-R560C protein can be detected at a low level in the retina [21]. Dark rearing *rd10* mutant mice delayed retinal degeneration for a week while dark rearing *rd1* mutant mice failed to rescue the loss of photoreceptor cells [20]. These animal models of PDE6B mutations have been providing an invaluable resource for understanding the molecular pathology and treatment of human RP patients [22,23,24]. 

Here, we report the identification of a new RD model, the *r28* mouse line, from a retinal phenotypic screen of N-ethyl-N-nitrosourea (ENU)-mutagenized mice. The *r28* mutant displays rapid photoreceptor cell death caused by a new *Pde6b* point mutation. We further explored therapeutic interventions in this RD mouse model by using a recombinant adeno-associated virus (rAAV) delivery system with subretinal injection in vivo. This work suggests that the *r28* mutant is a novel light-dependent RD mouse model; its photoreceptor loss can be rescued by dark rearing and the expression of the wild-type PDE6B protein, delivered by subretinally injected rAAV-hRho-wt-PDE6B.

## 2. Materials and Methods

### 2.1. Animals 

All experimental procedures were approved by the Animal Care and Use Committee (ACUC) at the University of California, Berkeley (Berkeley, CA, USA), and were conducted in accordance with the ARVO Statement for the Use of Animals in Ophthalmic and Vision Research. Mouse *r28* was identified from a mouse ENU-mutagenesis program published in previous papers [25,26]. Detailed procedures for the eye examination and mouse mating procedures were also published in our previous papers [26,27]. The mouse *r28* mutation was identified as a recessive retinal mutation and maintained in the C57BL/6J strain background. The phenotypes of homozygous mutant mice were examined and confirmed in many different litters at different generations since the *r28* mutant line was first identified and established over two decades ago.

### 2.2. Fundus Examination and Histology 

Fundus examination and retinal histological analysis were performed as previously described [26,27]. In brief, for fundus examination, mouse pupils were dilated with a mixed ophthalmic solution containing 0.5% atropine sulfate and 1.25% phenylephrine hydrochloride; abnormalities of retinal vessels and pigment distribution were examined by an indirect ophthalmoscope (Keeler) with a super 66 stereo Fundus lens. Fundus photos were taken with a Kowa Genesis fundus camera for small animals (Tokyo, Japan). For histology analysis, dissected eyes were fixed in a solution containing 2.5% glutaraldehyde and 2% paraformaldehyde in 0.1 M cacodylate buffer (pH 7.2) for at least 24 h at 4 °C; fixed eyes were post-fixed in 1% aqueous OsO_4_, stained *en bloc* with 2% aqueous uranyl acetate, and dehydrated through graded acetone; dehydrated samples were embedded in Epon 12-Araldyte 502 resin; 1 μm sections were cut and collected for staining with Toluidine blue; images were acquired with a Zeiss Axiovert 200 light microscope with a digital camera. For fundus examination, phenotypes were examined and confirmed in many different litters at different generations. For histology study, we examined at least three mice from one litter for these experiments, and the procedures were repeated on at least three litters of mice.

### 2.3. Electroretinography 

The procedure of whole-field scotopic electroretinography (ERG) was published previously [27]. In brief, mice raised in normal light environment were dark-adapted overnight; mice were anesthetized with a mixture of ketamine (100 μg per gram of body weight) and xylazine (16 μg per gram of body weight) in PBS; mouse pupils were dilated with a mixture of 1% atropine HCl and 2.5% phenylephrine HCl, and 0.5% proparacaine was used as a local anesthetic. A contact lens electrode was placed on the cornea with a layer of 2.5% hydroxypropyl methylcellulose (Gonisol; Novartis Ophthalmics, Duluth, GA, USA); differential electrodes were inserted in the cheeks while the grounding electrode was inserted into the tail. Light stimulation was performed at short intervals of low-intensity light followed by longer intervals of high-intensity light, from 0 cd.s/m^2^ to 31.62 cd.s/m^2^ in seven steps, and from 30 s intervals to 120 s intervals. At least three mice of each genotype were tested, and the averaged amplitude values of step 7 (31.62 cd.s/m^2^ light intensity) were presented. The ERG a-wave reflects the photoreceptor activity, and the b-wave reflects the inner retinal cell activity. ERG data were collected from 3-week-old heterozygous *r28*/*+* control, the homozygous *r28*/*r28* mutant, and dark-reared *r28*/*r28* mutant mice.

### 2.4. DNA Sequencing

Total RNA was prepared from retinas isolated from the homozygous *r28*/*r28* mutant mice using the TRIzol^®^ Reagent (Invitrogen Life Technologies, Carlsbad, CA, USA); cDNA was synthesized with the Superscript^TM^ First-Strand Synthesis System for RT-PCR kit (Invitrogen Life Technologies). The coding region of the *Pde6b* gene was PCR-amplified with the Platinum^®^ pfx DNA polymerase (Invitrogen Life Technologies) and sequenced. 

### 2.5. Western Blot Analysis 

The western blot analysis was performed as previously described [28]. A rabbit polyclonal anti-PDE6B antibody (Cat#PA1-722, Invitrogen) and a mouse monoclonal anti-α-tubulin (Cat# T9026, Sigma-Aldrich, Saint Louis, MO, USA) were used. We examined at least three mice from one litter for the experiment, and the analysis was repeated on at least three litters of mice.

### 2.6. Immunohistochemistry

Immunostaining of retinal frozen sections was performed as described previously [26], and all images were collected by a Zeiss LSM700 confocal microscope. A polyclonal PDE6B antibody (Cat#PA1-722, Invitrogen), monoclonal anti-Rhodopsin antibody (Cat#MAB5316, Chemicon International), and Alexa Fluor^TM^ 488 Phalloidin (Molecular Probes) were used for staining. We examined at least three mice from a litter for staining experiment, and the procedure was repeated on at least three litters of mice.

### 2.7. Generation of rAAV-hRho-PDE6B and Subretinal Injection

Wild-type or homozygous mutant *r28*/*r28 Pde6b* cDNA fragments covering the coding sequence were subcloned into an rAAV plasmid backbone pTR in which the original chicken β-actin promoter was replaced with the human rhodopsin promoter (hRho). To generate rAAV-PDE6B, the AAV transfer plasmids were constructed as follows: (1) cDNAs were synthesized from RNAs isolated from wild-type or *r28*/*r28* mouse retinas and used as templates for PCR amplification with a pair of primers (forward 5′-CAG GAA CAC CAT GAG CCT C and reverse 5′-TAG GAT ACA GCA GGT CGA GG). The cDNA fragments encoding the full length PED6B were subcloned into the pEGFP-N1 vector to create the pEGFP N1-wt or *r28* PDE6B plasmids; (2) AAV plasmid pTR-Rho was generated by replacing the smCBA promoter in the pTR-sbCBA-hGFP with the human rhodopsin promoter; (3) pEGFP N1-wt or *r28* PDE6B were used as templates to generate PCR fragments with a pair of primers (forward 5′-ATTTGCGGCCGCACCATGAGCCTCAGTGAG and reverse 5′-ATTTGCGGCCGCTTATAGGATACAGCAGGTCG), and the PCR fragments were digested with NotI and ligated with NotI digested and Antarctic Phosphatase-treated pTR-hRho backbone. The resulting plasmids pTR-hRho-wt-PDE6B or pTR-hRh-r28-PDE6B were sequence-confirmed and packaged into AAV5 to produce the rAAV-hRho-wt-PDE6B and rAAV-rRho-r28-PDE6B.

Subretinal injection was performed as previously described [29]. Mice at the age of postnatal day 12 were used for injection. The closed eye lids were cut open; ~1 μL of rAAV-hRho-wt-PDE6B (~4.7 × 10^12^ vg/mL) or rAAV-hRho-r28-PDE6B (~3.6 × 10^12^ vg/mL) were injected subretinally into the right eyes while the left eyes were left as un-inject controls. The mice were raised under normal light condition with a 12 h light/12 h dark cycle. Eyes were collected 6 weeks post-injection, when the mice were about 2 months old. For the AAV-hRho-wt-PDE6B, a total of nine mutant mice from three different litters, including both female and male mice, were used for the subretinal injection experiment. 

## 3. Results

### 3.1. Loss of Photoreceptor Cells in the r28 Mutant Mice

The *r28* mutation, identified from a screen of ENU-induced mice by fundus examination, displayed recessive retinal degeneration. The homozygous *r28*/*r28* mice developed rapid photoreceptor cell loss. At the age of postnatal day 21 (P21), compared to age-matched wild-type control (+/+) mice, mutant mice displayed abnormal fundus and attenuated retinal artery vessels (Figure 1A). Histology analysis revealed a loss of photoreceptor cells in the *r28* homozygous mutant retina. At P10, when the mouse eye was still closed, there was no obvious loss of photoreceptor cells observed in the *r28* homozygous mutant retina (Figure 1B). At the age of P21, however, the *r28* mutant retina showed an extensive loss of photoreceptor cells (Figure 1C); the wild-type control retina had 9–11 layers of photoreceptor nuclei in the outer nuclear layer (ONL), while the *r28*/*r28* retina only had 2–4 layers of photoreceptor cells left in the ONL. Thus, the *r28* mutation causes a rapid loss of photoreceptor cells in the homozygous mutant retina.

### 3.2. Dark Rearing Rescues Photoreceptor Cell Loss in the r28 Mutant Mice 

The retinal histology data revealed a rapid loss of photoreceptor cell layers in the *r28*/*r28* mice at weaning age (P21), but no loss was observed at P10, the young age at which the mouse eyes were closed. This indicates that the *r28* mutant phenotype may be light-dependent. Thus, we further investigated photoreceptor cell loss in *r28* mutant mice raised in a dark environment. The dark-reared homozygous *r28*/*r28* mutant mice displayed retinal histology comparable to the heterozygous *r28*/*+* control (Figure 2A); no extensive loss of photoreceptor cells was observed in the dark-reared *r28*/*r28* mutant mice, while the normal light-reared mutant *r28*/*r28* mice showed an obvious loss of photoreceptors (Figure 2A). Counting the number of photoreceptor nuclear layers–outer nuclei layers (ONL) (Figure 2B), control *r28*/*+* retinas displayed ~11 layers of cells; the dark-reared mutant *r28*/*r28* mice had ~9 layers of photoreceptor cells (*p* < 0.01, Student’s *t*-test between dark-reared *r28*/*r28* and *r28*/*+*); the normal light-reared mutant *r28*/*r28* mice only had ~4 layers of cells left (*p* < 0.001, Student’s *t*-test between *r28*/*r28* and *r28*/*+*). The ONL counts between dark-reared and light-reared *r28*/*28* mice are statistically significant (*p* < 0.001, *n* = 10). We further tested the visual function of *r28* mutant mice by scotopic ERG recording. Compared to the control *r28*/*+* mice, the regular light-reared mutant mice displayed significantly reduced ERG a- and b-waves; however, the reduction of a- and b-waves in *r28*/*r28* mutant mice was rescued by dark rearing (Figure 2C). Thus, our results indicate that the *r28* mutant phenotype is light-dependent, and dark rearing can rescue the visual function loss of the *r28* mutant mice.

### 3.3. A Pde6b Point Mutation Leads to Mutant PDE6B-T592I Protein in the r28 Mutant Mice

Sequence analysis was performed to identify the causative gene mutation that caused the phenotype in the *r28* mutant mice. Because of the apparent retinal degeneration phenotype, we sequenced the *Pde6b* gene using the cDNA made from the *r28*/*r28* mutant retina. Compared to the sequence encoding the 856 amino acid residues of mouse PDE6B protein, we detected a single nucleotide change (C to T) in the *r28* mutant, c.1775C>T for a p.T592I, which would result in a change of the 592nd amino acid residue of PDE6B protein from threonine (T, encoded by ACA) to Isoleucine (I, encoded by ATA). Western blot analysis was performed to examine the expression of mutant PDE6B-T592I protein. The mutant protein was detected in the retinal protein homogenates of P7 and P10 *r28*/*r28* mutant mice but, compared to normal PDE6B protein expression level in the wild-type control retinas, the mutant protein showed very low expression. The PDE6B-T592I expression was further studied by the immunostaining of P12 and P21 retinal frozen sections. In the wild-type retinal sections, both P12 (Figure 3C) and P21 (Figure 3D) sections showed strong PDE6B staining signals predominantly in the photoreceptor outer segments, but the PDE6B proteins were hardly detected in the *r28*/*r28* mutant retinas. Thus, the mutant PDE6B-T592I protein had a very low expression level in the *r28* mutant retina. 

### 3.4. Subretinal Injection of rAAV-hRho-wt-PDE6B Rescues Photoreceptor Cell Loss in the r28 Mutant Mice

To explore therapeutic intervention to prevent photoreceptor loss and to test whether expressing the normal wild-type PDE6B protein in the mutant *r28*/*r28* mice can rescue the RD phenotype, we used a gene-therapeutic approach mediated by recombinant adeno-associated virus (rAAV). Only one subretinal injection of rAAV-hRho-wt-PDE6B, expressing normal PDE6B protein under the control of a human rhodopsin promoter, can prevent the loss of photoreceptor cells in the *r28*/*r28* mutant mice (Figure 4). The P12 *r28*/*r28* mice (eyes were still closed) were subretinally injected with the rAAV-hRho-wt-PDE6B; injected mice were raised in a normal light environment for 6 more weeks, and eye samples were collected when the mice were 2 months old. A total of nine mutant *r28*/*r28* mice from three different litters, including both female and male mice, were used for the subretinal injection experiment. Retinal frozen sections were prepared for immunostaining with an anti-rhodopsin antibody and DAPI. Confocal images were taken to generate montaged images of immunostained retinas (Figure 4). All the injected eyes of these nine homozygous mutant *r28*/*r28* mice showed various degrees of rescue of photoreceptor cell loss. In the eyes of mutant *r28*/*r28* mice without injection, the retinas only had one layer of ONL left (Figure 4A). The montaged retina staining images from two *r28*/*r28* mice treated with rAAV-wt-PDE6B are shown (Figure 4B,C). In the first 2-month-old *r28*/*r28* mouse (Figure 4B, mouse#1) injected with rAAV-wt-PDE6B, one half of the retina displayed 4–8 layers of photoreceptor cells (Figure 4B, the right half; Figure 5C), suggesting that rAAV-wt-PDE6B can rescue photoreceptor cell loss in the *r28*/*r28* mutant retina. Although the other half only had one layer of cells left, this is likely caused by limited diffusion of subretinally injected rAAVs; probably no viruses were penetrated in this area. In another 2-month-old *r28*/*r28* mouse (Figure 4C, mouse#2) injected with rAAV-wt-PDE6B, a modest rescue was observed, as the whole retinal montage image displayed 4–7 layers of photoreceptor cells (Figure 4C and Figure 5D). As a control, mutant rAAV-hRho-r28-PDE6B was also injected into *r28*/*r28* littermates, and no rescue was observed in any mice injected (unpublished observation). Thus, the subretinal injection of rAAV expressing normal wild-type PDE6B can prevent photoreceptor cell death in the *r28* mutant retina.

### 3.5. The T592I Mutation in the r28 Mutant Mice Disrupts the Hydrogen Bond Formation between T592-Y524 and T592-Y645

The 3D structure of PDE6B is predicted by the SWISS-MODEL (https://swissmodel.expasy.org (accessed on 28 September 2023)) (Figure 6A). The intramolecular interactions of selected amino acid residues were analyzed with UCSF ChimeraX (https://www.rbvi.ucsf.edu/chimerax (accessed on 28 September 2023)) [30]. In the normal wild-type PDE6B protein, the T592 amino acid residue is predicted to form hydrogen bonds with the following amino acid residues: Tyrosine (Tyr or Y) 524 (Figure 6B), Phenylalanine (Phe or F) 588, Cysteine (Cys or C) 596, and Tyr (Y) 645 (Figure 6B). In the mutant PDE6B-T592I protein, the mutated I592 is predicted to only form hydrogen bonds with Phe588 and Cys596, but no hydrogen bonds can be formed between I592 and Y524 (Figure 6C) and between I592 and Y645 (Figure 6C). The disruption of hydrogen bonds in the PDE6B-T592I protein would affect the local packing near T592 caused by both size change and the break of hydrogen bonds, which could result in unstable mutant proteins with a reduced protein level and impaired protein function, thus causing the RD phenotype.

## 4. Discussion

In this study, we have identified and characterized a mouse *Pde6b* point mutation, a c.1775C>T for p.T592I substitution, resulting in the mutant PDE6B-T592I protein in the *r28* mouse line. PDE6B was the first gene identified with a causative mutation for human RP [7,8]. The PDE6B gene encodes rod cGMP-phosphodiesterase 6 beta, a protein subunit of the PDE6 complex localized on the rod outer segments that plays an essential role in the phototransduction cascade for sensing light in photoreceptor cells [31,32]. 

The homozygous *r28* mutant mice developed rapid photoreceptor cell loss, caused by a point mutation of the *Pde6b* gene that resulted in the mutant PDE6B-T592I protein. The PDE6B-T592I point mutation in the *r28* mutant mice causes recessive and light-dependent retinal degeneration. Like the *rd10* mice caused by the PDE6B-R560C point mutation [20], the loss of photoreceptors in the *r28* mutant mice can be prevented by dark rearing. This indicates that photoreceptor cell death is triggered and/or accelerated by light in the *r28* mutant mouse line. The mutant PDE6B-T592I proteins, although expressing at a low level, are sufficient for the survival of rod photoreceptor cells from dark-reared mice. The low expression of PDE6B-T592I protein is probably caused by the disruption of hydrogen bonds between T592–Y524 and T592–Y645 in the mutant protein. Intramolecular hydrogen bonds significantly contribute to protein stability and are necessary for maintaining protein structure [33]; tyrosine hydrogen bonds have been reported to make a large contribution to protein stability [34]. In the *r28* mutant mice, disrupted hydrogen bond formation between T592 and two tyrosine residues (Y524 and Y645) likely affects the stability of mutant PDE6B-T592I protein, and the reduced protein stability leads to very low protein expression. 

However, the underlying mechanism for the way mutated genes cause the death of photoreceptor cells is not well understood. The PDE6B-R560C mutation in the *rd10* mouse causes reduced stability and mislocalization of the PDE6B protein, and promotes cell death through increased ion influx [21]. The cGMP levels are lower in dark-reared *rd10* mice than in normal light-reared mutant mice [21]; thus, the hypoactive PDE6B protein in the *rd10* mice is still capable of cleaving cGMP. A recent study of dark-reared *rd10* mice showed rapid photoreceptor degeneration with short exposure to room light during in-vivo retinal imaging [35], with uncertain underlying mechanisms. Like the *rd10* mutation, it is likely that the PDE6B-T592I mutation in the *r28* mouse results in a hypoactive PDE6B mutant protein, which is still capable of cleaving cGMP.

Currently, certain gene-based therapeutic approaches show some promises to effectively treat recessive RP in animals and humans [32,36,37,38,39,40,41,42]. Genetic animal models of PDE6B mutations are important for understanding the molecular pathology and treatment of human RP patients. In this study, loss of photoreceptors was partially rescued in the *r28* mutant mice by just one injection of rAAV-wt-PDE6B to the mice at P12, when the eyes were still closed. Perhaps, PDE6B mutations with hypoactivity can be considered to undergo a therapeutic approach by subretinal injection of rAAV expressing normal wild-type PDE6B. The result confirms our hypothesis that the instability and low expression of PDE6B-T592I protein result in PDE6B hypoactivity, leading to the RD phenotype in the *r28* mutant mice. A subretinal injection of rAAV-mutant PDE6B-T592I failed to rescue photoreceptor loss in the *r28* mutant mice, suggesting that the mutant protein alone is not sufficient to protect the photoreceptors; this further confirms that the PDE6B-T592I mutant protein is hypoactive in this *r28* RD mouse model. 

In conclusion, the *r28* mutant line is a new RD mouse model for human RP. The initial gene therapy results using rAAV-wt-PDE6B support the feasibility and possibility of treating human RP patients who are associated with hypoactive PDE6B proteins in the photoreceptor cells. 

## Figures and Tables

**Figure 1 biomedicines-11-03173-f001:**
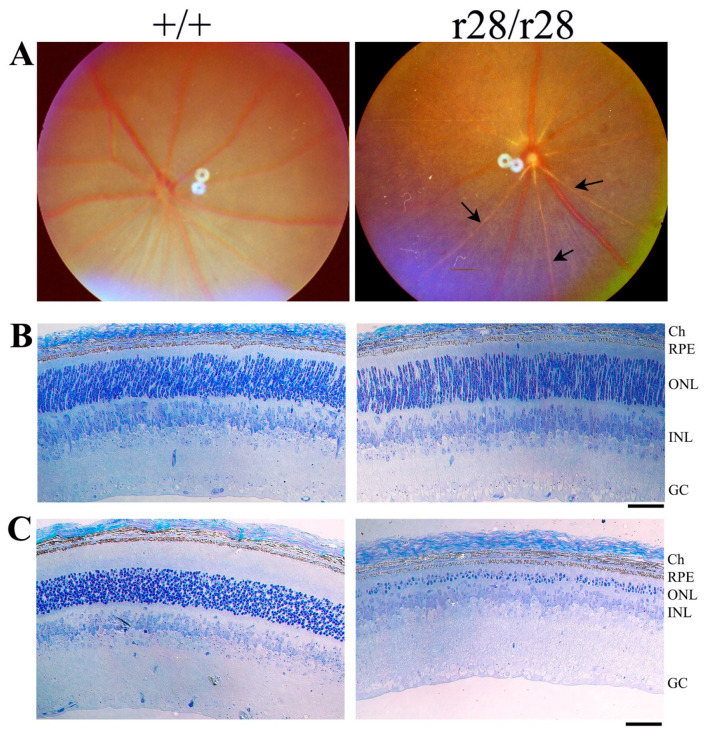
Loss of photoreceptor cells in the *r28* mutant mice. (**A**) Fundus photos of postnatal 21 days (P21) *r28* homozygous (*r28*/*r28*) and age-matched control wild-type (+/+) eyes. Compared to the control eye, the *r28* mutant retina displays attenuated artery vessels (arrows). (**B**) Retinal histology of *r28*/*r28* mutant and wild-type control at postnatal day 10 (P10). No loss of photoreceptor cells is observed in the *r28* homozygous mutant retina. (**C**) Retinal histology of *r28*/*r28* mutant and wild-type control mice at the age of P21. The *r28* mutant retina only has ~3–4 layers of photoreceptor cell nuclei left in the outer nuclear layer (ONL), while the control retina displays 9–11 layers of photoreceptor nuclei in the ONL. Ch, choroid; RPE, retinal pigment epithelial; ONL, outer nuclear layer; INL, inner nuclear layer; GC, ganglion cell layer. Scale bars, 50 μm.

**Figure 2 biomedicines-11-03173-f002:**
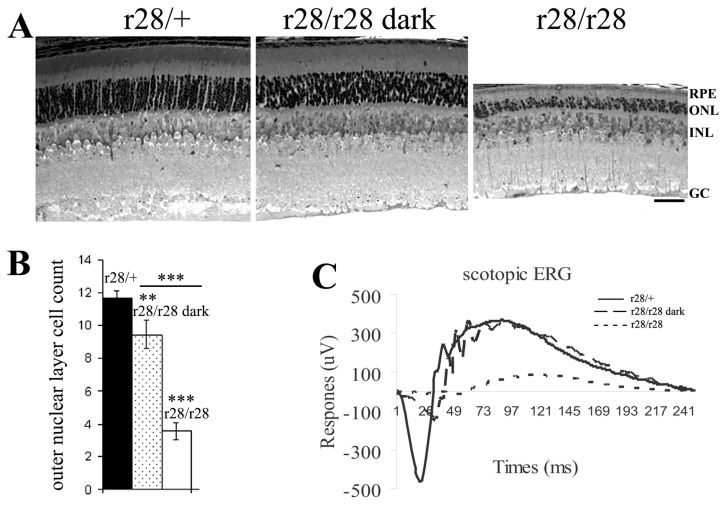
Dark rearing rescues photoreceptor cell loss in *r28* homozygous mutant mice. (**A**) Retinal histology comparison of P21 *r28* homozygous (*r28*/*r28*) mutant mice raised in a dark environment or regular light conditions. The retinal section from a regular light-reared *r28*/*r28* mouse (the right image) displays 3–4 layers of photoreceptor cell nuclei (ONL); in the retina of a mouse reared in a dark environment (the middle image), the loss of photoreceptor cells is partly rescued as there are ~9 layers of ONL observed. The *r28* heterozygous (*r28*/*+*) retinal section is shown as a control. Scale bar, 50 μm. (**B**) At P21, the control *r28*/*+* mice (*n* = 3) display ~11 layers of photoreceptor cells. The regular light-reared *r28*/*r28* mice (*n* = 10) have ~4 layers of cells left, while the dark-reared *r28*/*r28* mice (*n* = 10) still have ~9 layers of photoreceptor cells. Student’s *t*-test was used for statistical analysis, ** *p* < 0.01, *** *p* < 0.001. (**C**) ERG recording shows rescued a- and b-waves in the dark-reared *r28*/*r28* mice compared to the regular light-reared mutant mice.

**Figure 3 biomedicines-11-03173-f003:**
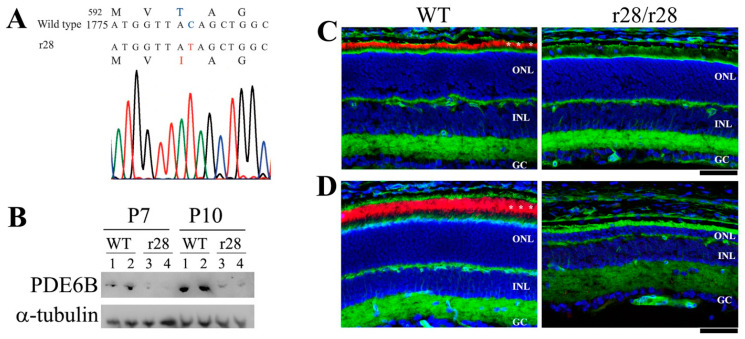
A point mutation in the *Pde6b* gene leads to mutant PDE6B-T592I protein in the *r28* mutant mice. (**A**) In the *r28* mutant mice, a point mutation c.1775C>T in exon 14 of the *Pde6b* gene results in a substitution of the 592 amino acid residue of PDE6B protein from threonine (T) to isoleucine (I). (**B**) Western blot analysis to examine the mutant PDE6B-T592I proteins in the *r28* mutant retinas. The mutant proteins are detected in the retinas of P7 and P10 homozygous *r28* mice; however, the mutant protein levels are significantly reduced compared to normal PDE6B proteins in the control wild-type retinas. Western images from retinal samples of two wild-type (1, 2) and two mutant (3, 4) mice at P7 and P10 are shown. (**C**,**D**) Immunostaining of *r28* mutant retinal frozen sections with an anti-PED6B antibody (red), phalloidin (green), and DAPI (blue). Compared to the wild-type sections from mice at both P12 (**C**) and P21 (**D**), which display strong PDE6B signals in the photoreceptor outer segment (indicated by asterisks), the PDE6B proteins are hardly detected in the *r28* homozygous retinas. ONL, outer nuclear layer; INL, inner nuclear layer; GC, ganglion cell layer. Scale bars, 50 μm.

**Figure 4 biomedicines-11-03173-f004:**
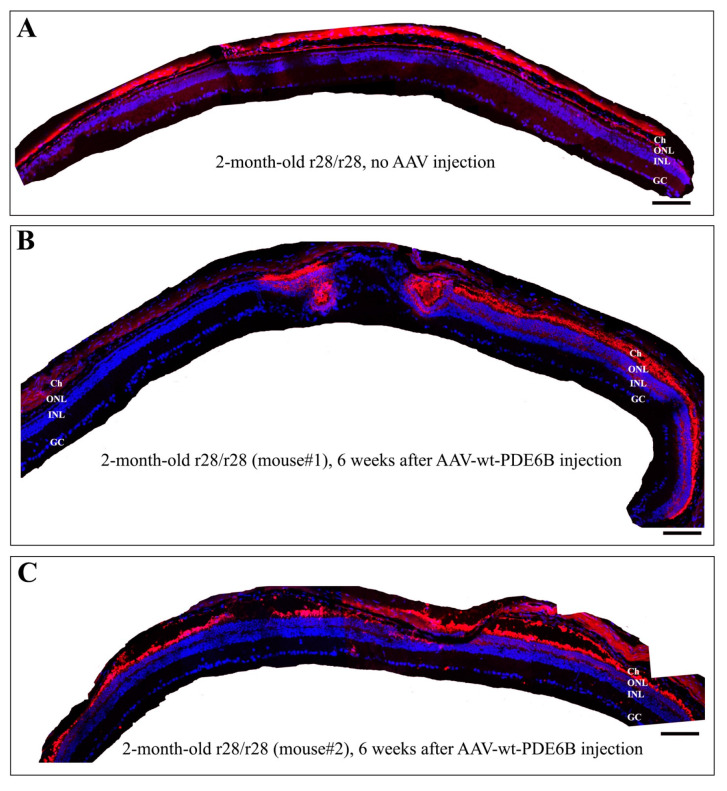
Subretinal injected rAAV expressing normal wild-type PDE6B can rescue photoreceptor cell loss in the *r28* mutant retina. The rAAV-hRho-wt-PDE6B was injected into P12 *r28*/*r28* littermates. About 6 weeks post-injection, these injected eyes were collected for immunostaining with an anti-rhodopsin antibody (red) and DAPI (blue). Montaged images show immunostained retinas from one untreated control and two AAV-wt-PDE6B-treated *r28*/*r28* mice. (**A**) A 2-month-old *r28*/*r28* mouse without AAV injection only has ~1 layer of photoreceptor cells left. (**B**) In one 2-month-old *r28*/*r28* mouse (mouse#1) injected with rAAV-wt-PDE6B, one half of the retina displays 4–8 layers of photoreceptor cells, and the other half only has one layer of cells left. (**C**) In another 2-month-old *r28*/*r28* mouse (mouse#2) injected with AAV-wt-PDE6B, a modest rescue is observed, as the whole retinal areas displays 4–7 layers of photoreceptor cells. Ch, choroid; ONL, outer nuclear layer; INL, inner nuclear layer; GC, ganglion cell layer. Schedule bars, 100 μm.

**Figure 5 biomedicines-11-03173-f005:**
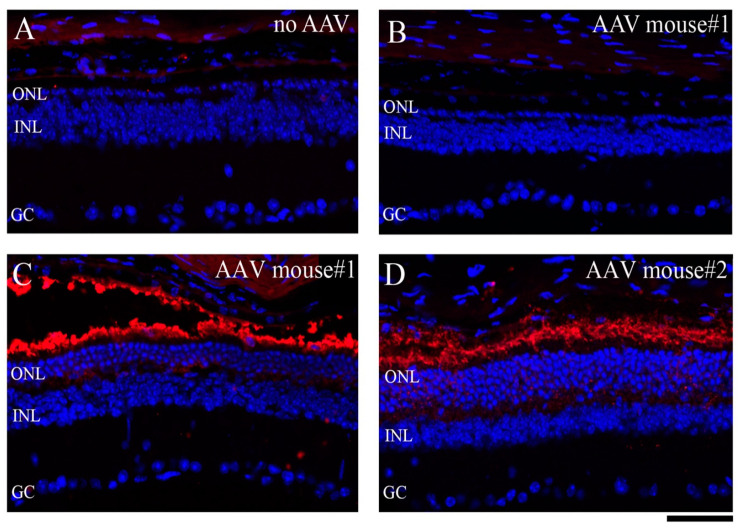
Enlarged immunostaining images of rAAV-wt-PDE6B-injected retinas of 2-month-old *r28*/*r28* mice in Figure 4. Photoreceptor out-segments were immunostained with an anti-rhodopsin antibody (red) and cell nuclei were labeled by DAPI (blue). (**A**) A retinal section of the un-injected left eye of mouse #1 displays only one single layer of photoreceptor cells and no rhodopsin staining signal. (**B**,**C**) Retinal sections of the injected right eye of mouse #1. (**B**) No rhodopsin staining, and no photoreceptor rescue were observed. This enlarged image was taken from the left one-half retina of mouse #1 injected with rAAV-wt-PDE6B in Figure 4B. (**C**) Enlarged image of the right region with the rescued ONL of the retina of mouse #1 in Figure 4B shows rhodopsin staining in these rescued photoreceptor cells; there are ~4 layers of outer nuclei left, compared to one layer in un-rescued retinas (images (**A**,**B**)). (**D**) A retinal section of the injected eye of mouse #2 displays rhodopsin staining and ~5–7 layers of rescued photoreceptor cells. ONL, outer nuclear layer; INL, inner nuclear layer; GC, ganglion cell layer. Scale bars, 50 μm.

**Figure 6 biomedicines-11-03173-f006:**
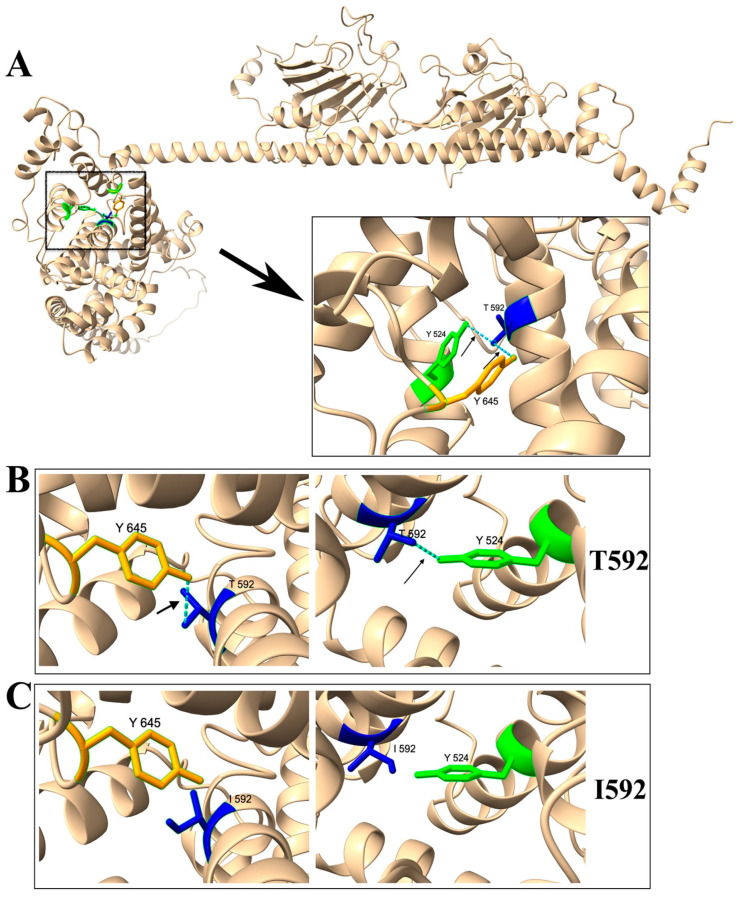
Three-dimensional protein structure modeling predicts that T592I mutation disrupts the hydrogen bond formation between T592-Y645 and T592-Y524 in PDE6B. (**A**) The 3D structure of mouse PDE6B protein predicted by the SWISS-MODEL (https://swissmodel.expasy.org (accessed on 28 September 2023)) is shown. The enlarged box displays an area showing the intramolecular interactions of amino acid residues T592, Y524, and Y645 using the UCSF ChimeraX-1.6.1. (https://www.rbvi.ucsf.edu/chimerax (accessed on 28 September 2023)). (**B**) In the normal wild-type PDE6B, the T592 residue forms a hydrogen bond with Y645 as well as with Y524. (**C**) In the mutant PDE6B, the I592 residue cannot form a hydrogen bond with Y645 or Y524.

## Data Availability

The data presented in this study are available on request from the corresponding author.

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
