# Peer review of "Identification and Characterization of Retinitis Pigmentosa in a Novel Mouse Model Caused by PDE6B-T592I"

_biomedicines, 2023, doi:10.3390/biomedicines11123173_

Round 1

Reviewer 1 Report

Comments and Suggestions for Authors

The authors have written an extremely fascinating and highly scientific article.

The methods are highly repeatable and have sufficient description

The results are described in detail and supported by figures and graphs

The article is appropriate for the journal, and I advise that it be approved for publication following a few minor English language corrections and a rephrasing of the study's goal in the abstract and introduction.

Comments on the Quality of English Language

a few minor English language corrections

Author Response

Reviewer#1
1)    Minor editing of English 
2)    Comments: The authors have written an extremely fascinating and highly scientific article; The methods are highly repeatable and have sufficient description; The results are described in detail and supported by figures and graphs; The article is appropriate for the journal, and I advise that it be approved for publication following a few minor English language corrections and a rephrasing of the study's goal in the abstract and introduction.
•  We thank the reviewer’s supportive comments. We have revised the abstract and the introduction to ensure correct English grammar and appropriate descriptions.  

Reviewer 2 Report

Comments and Suggestions for Authors

The aim of this manuscript is to investigate a mouse model of retinitis pigmentosa (RP). The author developed a point mutation of the Pde6b gene and observes that homozygous mice develop RP. This condition can be rescued through dark rearing or the introduction of AAV-wildtype PED6B.

The manuscript is interesting  with a well-structured design and presenting significant results. However, there are a couple of points that require further attention:

  1. The rationale behind choosing T592I as the target needs more thorough justification.
  2. It has been noted by others that W378 or R799 mutations are also implicated in RP development. Please provide clarification on this aspect.

Author Response

Reviewer#2

The aim of this manuscript is to investigate a mouse model of retinitis pigmentosa (RP). The author developed a point mutation of the Pde6b gene and observes that homozygous mice develop RP. This condition can be rescued through dark rearing or the introduction of AAV-wildtype PED6B.

The manuscript is interesting with a well-structured design and presenting significant results. However, there are a couple of points that require further attention:

  1. The rationale behind choosing T592I as the target needs more thorough justification.
  • We did not intend to generate the T592I mutation. This mutant line is a random mutation that we identified from a screen of ENU-induced mutant mice based on the retinal phenotypes. We then identified the causative gene mutation as Pde6b-T592I and characterized its unique retinal phenotypes.

  1. It has been noted by others that W378 or R799 mutations are also implicated in RP development. Please provide clarification on this aspect.
  • The W378X and R799X mutations are human PDE6B gene mutations that cause RP phenotypes. Compound heterozygosity of these two PDE6B mutations could also lead to RP. Both mutations lead to premature stop codon, resulted in truncated PDE6B proteins. These mutations should be counted in “https://sph.uth.edu/retnet” described in the first paragraph of the introduction.

Reviewer 3 Report

Comments and Suggestions for Authors

The article by Xia, C.-H. et al. propose a very interesting experiment describing a new model of retinitis pigmentosa. It appears scientifically correct and well organized. I think the text has various issues, but I hope all of them can be solved.

I can´t find significant language issues.

I have some major comments to make:

- Introduction. I feel this section is limited. The second paragraph (lines 52-68) includes information that can certainly be categorized as results. You should also explain the ENU-mutagenized mice. In addition, some text of the discussion may better fit in introduction. I would like to see an enhancement here.

- Materials and Methods. You should mention the exact number of animals (section 2.1), then explain the ages of their sacrifice. In results, section 3.3, you mention P7 and P12, in addition P10 and P21 were mentioned before. You only mention 9 mice in section 2.7 describing the adenovirus experiment (also later in section 3.4). A complete new paragraph and a table should be included here to clearly explain the methods.

- Results, line 151. You should mention how you measured photoreceptor cell loss (I mean, the technique). Reading the remaining paragraph it seems you measured the number of layers, but not the actual photoreceptor cells, so the final sentence of this paragraph (160) seems inappropriate. May you properly measure the number of cells? The photo shows an intense loss of cells, but I would like a proper measure of it, as long as the number of layers is only an indirect method of measure. Some options may be image analysis software or directly counting cells in 40x fields. Currently, the use of cellular layers to measure actual cellularity is a limitation in the study. All the text should be refined to reflect this point.

There are also various minor comments:

- In the text you mention the age of 21 days (P21) and three weeks. I think it would be easier if you employ the same time reference in all the text. It is probably better to express the age in days (an example in section 3.2 and figure 2).

- Section 2.3. You should (briefly) mention what are a and b-waves, mentioned in section 3.2 and figure 2.

- Section 2.7. May be AAV-hRho-wt-Pde6b is not properly written.

- Section 3.4, lines 237-238. This first sentence seems more appropriate in discussion.

- Section 3.4, line 245. (…) mice, whose eyes were still closed, were (...). Please include a period before “injected”; the sentence is too long and difficult. I think in line 255 the correct form is “are shown”.

- Section 3.4, lines 263-266. Here you mention you injected the same adenovirus as a control. This is badly explained. “Mutant rAAV-PDE6B-T594I” is not mentioned anywhere in the text. According to section 2.7, may be “rAAV-hRho-r28-PDE6B

- Figure 4. Why did you manually cropped these images? They should have black background. You already explained the B image in the text, please do not be redundant in the legend. Please include in the legend several (very brief) references to indicate that photoreceptors are stained in red with rhodopsin (this way it will be easier to be understood, this also applies to figure 3). Figures 4 and 5 are a bit confuse: I suggest merging them to be easier to understand (you can include images of figure 4 as an small insert for images in figure 5).

- To enhance the introduction section, you can move a lot of information in the first two paragraphs of discussion, describing the rd1 and rd10 mutant mice.

- Discussion, lines 351-354. Please, revise the logic of the commas in this (long) sentence.

Author Response

Reviewer#3

 The article by Xia, C.-H. et al. propose a very interesting experiment describing a new model of retinitis pigmentosa. It appears scientifically correct and well organized. I think the text has various issues, but I hope all of them can be solved. 

I can´t find significant language issues.

 - Introduction. I feel this section is limited. The second paragraph (lines 52-68) includes information that can certainly be categorized as results. You should also explain the ENU-mutagenized mice. In addition, some text of the discussion may better fit in introduction. I would like to see an enhancement here.

  • We thank the reviewer’s comments. The Introduction has been substantially revised. The second paragraph (lines 52-68) has been revised. The generation of ENU-mutagenized mice has been reported in many previous papers, we cited a few of these papers in the 2.1.

 - Materials and Methods. You should mention the exact number of animals (section 2.1), then explain the ages of their sacrifice. In results, section 3.3, you mention P7 and P12, in addition P10 and P21 were mentioned before. You only mention 9 mice in section 2.7 describing the adenovirus experiment (also later in section 3.4). A complete new paragraph and a table should be included here to clearly explain the methods.

  • We added the number of animals used for each experiment in methods. Mouse number used was described in experimental procedures or in the figure legends.

- Results, line 151. You should mention how you measured photoreceptor cell loss (I mean, the technique). Reading the remaining paragraph it seems you measured the number of layers, but not the actual photoreceptor cells, so the final sentence of this paragraph (160) seems inappropriate. May you properly measure the number of cells? The photo shows an intense loss of cells, but I would like a proper measure of it, as long as the number of layers is only an indirect method of measure. Some options may be image analysis software or directly counting cells in 40x fields. Currently, the use of cellular layers to measure actual cellularity is a limitation in the study. All the text should be refined to reflect this point.

  • It has been well documented and accepted in the retinal research field that either the number of the outer nuclear layer (ONL) or the thickness of the ONL, rather than the number of the outer nuclei, can be used to describe the loss of rod photoreceptor cells in typical RD models. It’s a common practice to measure the number of outer nuclear layers unless the organization of ONL is disrupted, which is not the case in this RD model. Therefore, we follow the common practice in the field and present the data consistent with other RD mutants reported previously. The outer nuclear layer contains the photoreceptor cell nuclei, which displays 9-11 layers in the wild-type retina. The r28 mutant retina shows an obvious loss of photoreceptor cell layers. We disagree with the necessity to count the nuclei numbers of ONL.

There are also various minor comments:

- In the text you mention the age of 21 days (P21) and three weeks. I think it would be easier if you employ the same time reference in all the text. It is probably better to express the age in days (an example in section 3.2 and figure 2).

  • We have changed three weeks to P21.

- Section 2.3. You should (briefly) mention what are a and b-waves, mentioned in section 3.2 and figure 2.

  • We have added “The ERG a-wave reflects the photoreceptor activity, and the b-wave reflects the inner retinal cell activity” to Section 2.3.

 - Section 2.7. May be AAV-hRho-wt-Pde6b is not properly written.

  • Corrected as AAV-hRho-wt-PDE6B.

- Section 3.4, lines 237-238. This first sentence seems more appropriate in discussion.

  • The sentence has been moved to the discussion.

- Section 3.4, line 245. (…) mice, whose eyes were still closed, were (...). Please include a period before “injected”; the sentence is too long and difficult. I think in line 255 the correct form is “are shown”.

  • We’ve changed the sentence to “The P12 r28/r28 mice (eyes were still closed) were subretinally injected with the rAAV-hRho-wt-PDE6B”. The original line 255 has been changed to “are shown”.

- Section 3.4, lines 263-266. Here you mention you injected the same adenovirus as a control. This is badly explained. “Mutant rAAV-PDE6B-T594I” is not mentioned anywhere in the text. According to section 2.7, may be “rAAV-hRho-r28-PDE6B

  • We have revised to “rAAV-hRho-r28-PDE6B”.

- Figure 4. Why did you manually cropped these images? They should have black background.

  • Figure 4 shows montaged images combined from many separated high-resolution images to give a view of the whole retina. It is important to show that the rescued rods occurred around the injected AAV site, since injected AAV could not spread into the whole retina. We added these sentences in the revised manuscript.

 You already explained the B image in the text, please do not be redundant in the legend. Please include in the legend several (very brief) references to indicate that photoreceptors are stained in red with rhodopsin (this way it will be easier to be understood, this also applies to figure 3).

  • Figure 4B legend has been revised. In the Figure 4 legend, it is stated “…these injected eyes were collected for immunostaining with an anti-rhodopsin antibody (red) and DAPI (blue).” For Figure 3 legend for C and D, it is stated “(C-D) Immunostaining of r28 mutant retinal frozen sections with an anti-PED6B antibody (red), phalloidin (green) and DAPI (blue)”.

Figures 4 and 5 are a bit confuse: I suggest merging them to be easier to understand (you can include images of figure 4 as an small insert for images in figure 5).

  • We do not prefer to make the images of both figures too small to the readers. Figure 4 shows macroscopic information from retinas of three mice, Figure 5 shows a microscopic view.

- To enhance the introduction section, you can move a lot of information in the first two paragraphs of discussion, describing the rd1 and rd10 mutant mice.

  • The introduction has been extensively revised to provide necessary information about the previous findings of rd1 and rd10 models.

 - Discussion, lines 351-354. Please, revise the logic of the commas in this (long) sentence.

  • This long sentence has been revised to two sentences.

Round 2

Reviewer 3 Report

Comments and Suggestions for Authors

The manuscript by Xia, C.-H. et al. was improved, according to reviewer´s suggestions. I can accept some of the authors rebuttals (for example the one about merging figures 4 and 5 to ease the understanding of both), while other rebuttals are more difficult to accept (the manual crop in figure 4 is a manipulation of the image). In any case, I won´t add anything about them.

However, two of the issues were not properly addressed:

- Materials and Methods. What I meant with this major issue (was the second in my previous review) was a mention of the exact number of animals used in all the experimental procedures (in section 2.1 – Animals). You already mention in most of the following subsections the use of at least 3 litters… but how much animals? Did you employ the same litters for different experiments? If you used several batches or litters for each experiment, you should clearly explain this point. A table may be included here to clearly explain the methods, but, in any case, they should be clear (at least) in section 2.1. The number of studied subjects is a basic description in materials and methods.

- Results, line 172. With this previous third major issue I didn´t wanted to mean that the cell layer measure is not valid, but it is a (small) limitation in the measure of cellular loss. I understand that a proper measure of the number of cells is difficult and won´t add significant information to the study, but sentence in line 181 should be changed to “rapid loss of photoreceptor cell layers”. This is what you are actually measuring. All the text should be refined to reflect this point (for example, in line 345). However, I can accept statements like “loss of rod photoreceptors” or even “loss of photoreceptor cells” at some points: the cell loss is evident in the images, but what you properly quantified is cellular layer loss.

I can´t find significant language issues.

There is a pair of minor comments:

- Introduction, line 38. That mostly affects.

- Figure 1 legend. Why do you remove (P10) after postnatal day 10? You just added P21 after postnatal 21 days in this legend…

Author Response

- Materials and Methods. What I meant with this major issue (was the second in my previous review) was a mention of the exact number of animals used in all the experimental procedures (in section 2.1 – Animals). You already mention in most of the following subsections the use of at least 3 litters… but how much animals? Did you employ the same litters for different experiments? If you used several batches or litters for each experiment, you should clearly explain this point. A table may be included here to clearly explain the methods, but, in any case, they should be clear (at least) in section 2.1. The number of studied subjects is a basic description in materials and methods.

  • We added the following sentence in the section 2.1: “The phenotypes of homozygous mutant mice were examined and confirmed in many different litters at different generations since the r28 mutant line was first identified and established over two decades ago”. We also revised the section 2.2. The mouse information provided in the manuscript is adequate to address the reproducible datasets of this work.

- Results, line 172. With this previous third major issue I didn´t wanted to mean that the cell layer measure is not valid, but it is a (small) limitation in the measure of cellular loss. I understand that a proper measure of the number of cells is difficult and won´t add significant information to the study, but sentence in line 181 should be changed to “rapid loss of photoreceptor cell layers”. This is what you are actually measuring. All the text should be refined to reflect this point (for example, in line 345). However, I can accept statements like “loss of rod photoreceptors” or even “loss of photoreceptor cells” at some points: the cell loss is evident in the images, but what you properly quantified is cellular layer loss.

  • We agree with the reviewer’s suggestion to use “rapid loss of photoreceptor cell layers” in the results section, and we have changed the first sentence for the result section 3.2. to “The retinal histology data revealed a rapid loss of photoreceptor cell layers in the r28/r28 mice at weaning age (P21)”. Based on the general knowledge in the literation for many PDE6B mutations cause the death of rod photoreceptors, it is appropriate to use these statements in abstract, introduction and discussion.
  •  

There is a pair of minor comments:

- Introduction, line 38. That mostly affects.

  • The sentence has been revised.

- Figure 1 legend. Why do you remove (P10) after postnatal day 10? You just added P21 after postnatal 21 days in this legend…

  • (P10) has been added.